# The Compartment and Variety Effects Jointly Shape Pummelo Endophytic Mycobiota

**DOI:** 10.3390/jof12010023

**Published:** 2025-12-27

**Authors:** Pingzhi Wu, Congyi Zhu, Zhu Yu, Chuanhong Ren, Zhengyan Fan, Ruimin Zhang, Pengtao Yue, Yongjing Huang, Guiming Deng, Jiwu Zeng

**Affiliations:** Institute of Fruit Tree Research, Guangdong Academy of Agricultural Sciences, Key Laboratory of South Subtropical Fruit Biology and Genetic Resource Utilization, Ministry of Agriculture and Rural Affairs, Guangdong Provincial Key Laboratory of Science and Technology Research on Fruit Tree, Guangzhou 510640, China

**Keywords:** pummelo, fungal community, next generation sequencing, phyllosphere, fruit tree

## Abstract

The plant microbiome plays important roles in plant growth and resistance, but its assembly and affecting factors have not been fully studied for most of the agricultural plants. In this study, the endophytic mycobiota of the leaves and roots and the rhizosphere soils of five pummelo varieties were profiled based on the amplicon sequencing of the fungal internal transcribed spacer (ITS). The fungal richness and diversity were significantly different among the compartments, but not among the pummelo varieties. The composition and structure of the endophytic mycobiota of the compartments were significantly different across all five pummelo varieties. These suggest that the variety effect is weaker than the compartment effect, but still significant in shaping the pummelo mycobiota. Specifically, the dominant leaf endophytic fungal taxa (e.g., *Fusarium* and *Zasmidium*), and the root selection of fungal genera from the rhizosphere soils, were significantly different among the varieties. And also, the variety effect is more significant in shaping the leaf endophytic mycobiota than those of the roots. Finally, the pummelo varieties also showed some consistent alterations on the endophytic mycobiota, such as the root enrichment of *Exophiala* species. Our study indicates that the endophytic mycobiota of pummelos is significantly and interactively affected by plant variety and compartment effects, and suggests some fungi of interest for further tests.

## 1. Introduction

Pummelos (*Citrus grandis* (L.) Osbeck), along with sweet oranges, tangerines, and mandarins, are among the most widely cultivated citrus fruits in the world [1]. In China, pummelos have been cultivated for about 4000 years and thus have accumulated very abundant germplasm resources [1,2]. For example, the well-known pummelo varieties ‘Guanxi’ pummelo, ‘Shatian’ pummelo, ‘Pingshan’ pummelo, and ‘Wendan’ pummelo are native to China. These varieties have been cultivated in South China for over hundreds of years [1]. In addition, some varieties from other countries, such as Thailand and Vietnam, have greatly enriched the pummelo market and germplasm resource in China [3].

During growth, the plant recruits and selects specific microbiomes, mainly mycobiome and bacteriome, to assemble and achieve success in its phyllosphere and rhizosphere compartments [4,5]. Of these, the endophytic microbiome can colonize the inner spaces of plant organs without causing apparent disease symptoms and also can escape the plant immune responses [6]. The selected fungal and bacterial species are shaped by plant genetics, plant growth and development stage, and plant compartment, and are crucial to plant fitness and resistance [4,7]. For example, the composition of both the plant phyllosphere and rhizosphere microbiome could be altered to improve their antagonistic species and resistance to plant pathogens [8,9]. Also, the root-selected bacterial species of *Rhizobiaceae* contributed largely to plant resistance to abiotic stresses such as drought [10]. These findings suggest that the plant microbiome can be harnessed for improving plant biotic and abiotic resistance through the cultivation and community synthesis of specific functional microbes [11]. However, the assembly, composition and influencing factors of the microbiome of many agricultural plants have not been well studied, which hinders the development of effective and rationally designed microbial communities for sustainable agriculture.

Plant variety and compartment are two important factors affecting the structure of the plant microbiome [12,13]. For example, the wilt disease-resistant and -sensitive tomato varieties showed different root microbial composition; the roots of the resistant variety enriched microbial taxa that contributed to the plant resistance to the soil-borne pathogen *Ralstonia solanacearum* [14]. The plant rhizosphere is a hot area of interactions between plant and soil microbes; the plant roots secrete abundant exudates to recruit microbial communities to live in the rhizosphere soils and the roots [4]. The phyllosphere microbial communities are generally different from those of the rhizosphere; many microbes of the phyllosphere plant organs are assembled from the air [15]. Many studies have investigated the plant variety and compartment effects on citrus species separately, and found that pummelos had a distinctive phyllosphere microbiota compared to that of sweet oranges, tangerines, and mandarins [16]. However, the variety effect and its relationships with other factors have not been thoroughly studied on citrus. In this study, the endophytic mycobiota of leaves, roots, and rhizosphere soils of five pummelo varieties were profiled. These varieties have important economic value and are wildly cultivated in South China. We aimed to explain the interactive effects of the variety and compartment with pummelo endophytic mycobiota with the following questions: (1) is the variety effect different among the compartments of leaf, root, and rhizosphere soil, and (2) is the compartment effect consistent across different varieties?

## 2. Materials and Methods

### 2.1. Pummelo Orchard and Sampling

The selected pummelo orchard, of about 5.3 ha, is located in Sifangjing village (112°22′1.175″ E, 24°49′22.228″ N), Baoan Town, Lianzhou City, Guangdong Province, China. The area has a central subtropical monsoon climate, the average annual air temperature is 19.7 °C, and the average annual precipitation is 1622 mm. The soil type is lateritic red soil [17]. The main cultivated varieties were ‘Hongxinmiyou’ (HX) which is a mutant of ‘Guanxi’ pummelo, its further improved mutant ‘Sanhongmiyou’ (SH), ‘Shatian’ pummelo (STY), and two varieties of green pummelo, ‘Siam Red Ruby’ pummelo (HBS) and Vietnamese green pummelo (YN). The root stock variety was Sour pummelo. These varieties were selected based on their important commercial value. HX and STY are widely cultivated in South China, SH, HBS, and YN are key varieties promoted by the local government. All varieties are natural breeding, not genetically modified. The pummelo trees were planted in 2015, while the scions of the varieties HBS and YN were re-grafted (via top-grafting) to the ‘Guanxi’ pummelos in 2019 and 2020, respectively. On 4 July 2024, six trees of each of the five pummelo varieties were randomly selected for sampling, which resulted in a total of 30 trees. For each tree, a spring leaf was picked from each of the four directions and the tree center, and the five leaves were mixed as a phyllosphere sample. The tree roots were dug out by a hoe and hand-shaken three to five times to remove loosely adhered soils. Then, the closely adhered soils were removed mechanically by a brush and collected as the rhizosphere soil sample. The fine roots, of diameter smaller than 2 mm, were collected as the root sample. The leaves and roots were surface-sterilized by soaking in 1 × TE buffer (Tris-EDTA buffer: 10 mM Tris, 1 mM EDTA, pH = 8.0, with 0.1% Triton X-100 added) for 30 s, 75% ethanol for 15 s, 2% bleach for 15 s, and sterile water three times [18], and then cut into small pieces. All the samples were stored under −80 °C overnight before DNA extraction.

### 2.2. DNA Extraction and Amplicon Sequencing

The leaves and roots were ground with liquid nitrogen. The total microbial DNA was extracted from 1 g of leaf and root powder and the rhizosphere soils, respectively. The extraction was processed following the protocol of the TGuide S96 Magnetic Soil/Stool DNA Kit (Tiangen Biotech Co., Ltd., Beijing, China) and monitored on 1.8% agarose gels. Then, the DNA was diluted to 1 ng/µL with sterile water, and a partial sequence of the fungal internal transcribed spacer (ITS rDNA) region was amplified with the primer set ITS1-1F (5′-CTTGGTCATTTAGAGGAAGTAA-3′) and ITS2 (5′-GCTGCGTTCTTCATCGATGC-3′) as described [19]. The polymerase chain reactions (PCRs) were processed with a 20 μL system (Toyobo Co., Ltd., Osaka, Japan): 3 µM of both primers, 5–50 ng template DNA, KOD FX Neo Buffer 5 μL, dNTP (2 mM each) 2 μL, KOD FX Neo 0.2 μL. The PCR program was set with initial denaturation at 95 °C for 5 min; 20 cycles of denaturation at 95 °C for 30 s, annealing at 50 °C for 30 s, and extension at 72 °C for 40 s; a final extension at 72 °C for 7 min. The PCR of each sample was conducted in triplicate, and the final products were pooled as a PCR mixture. The target DNA fragment was viewed by electrophoresis on 1.8% agarose gel and purified with Omega DNA purification kit (Omega Inc., Norcross, GA, USA). The purified DNA fragments were quantified using Qsep-400 (BiOptic, Inc., New Taipei, Taiwan, China) and used for library construction using TruSeq^®^ DNA PCR-Free Sample Preparation Kit (Illumina, San Diego, CA, USA). The paired-end sequencing (2 × 250) was processed on an Illumina novaseq6000 platform (Beijing Biomarker Technologies Co., Ltd., Beijing, China).

### 2.3. Bioinformatic Analysis

The raw reads were filtered by Trimmomatic version 0.33 based on the quality of single nucleotide [20], and their primer sequences were identified and removed by Cutadapt version 1.9.1 [21]. De-noising and the removal of chimeric sequences were processed by DADA2 [22] adapted in the QIIME2 2020.6 platform [23]. The amplicon sequence variants (ASVs) were conducted by DADA2, and ASV counts of less than 2 were filtered. Taxonomy annotation of the ASVs was performed based on the Naive Bayes classifier using UNITE v7.2 (https://unite.ut.ee/ (accessed on 12 September 2024)) as a reference database.

### 2.4. Statistical Analysis

The alpha diversity indices (Observed species, Shannon, Simpson, ACE, Chao1, Good’s coverage, and phylogenetic diversity) and the relative abundance of each taxon were calculated based on the counted ASVs in QIIME2 and R v4.0.0 software [24]. The comparisons of these indices and relative abundances among multiple groups were conducted with Kruskal–Wallis test. The estimations of fold change and significance of fungal taxa from the rhizosphere soils to the roots were performed in DESeq2 [25]. The beta diversity was calculated based on the binary Jaccard, Bray–Curtis, weighted Unifrac, and unweighted Unifrac distance matrices in the vegan package [26]. PERMANOVA was performed to compare the effects of compartment (phyllosphere leaf, root, and rhizosphere soil) and variety (HX, STY, HX, HBS, and YN) on the composition and structure of fungal communities using the adonis command in vegan package [26]. The Bray–Curtis distance matrix was subjected to principal coordinate analysis (PCoA) using the pcoa command in the Ape package [27], and the first two PCoAs were plotted using ggplot2 [28]. The data presented in a heatmap were analyzed and plotted in the package pheatmap [29].

## 3. Results

### 3.1. The Richness and Diversity of Pummelo Mycobiota

The fungal richness, represented by the observed number of fungal ASVs, of pummelo leaves, roots, and rhizosphere soils, was 149.3 ± 71.3 (mean ± standard deviation), 167.7 ± 103.9, and 217.8 ± 98.2, respectively (Figure 1A). The fungal diversity, represented by the Shannon diversity index, of pummelo leaves, roots, and rhizosphere soils, was 3.7 ± 0.5, 4.1 ± 0.9, and 4.6 ± 0.5, respectively (Figure 1B). Both of the indices were significantly different among the three plant compartments (*p* < 0.05), but were not significantly different among the five pummelo varieties across the three compartments (Figure 1).

The compartment effect on pummelo mycobiota was different among the varieties. For example, the fungal richness of YN (*p* < 0.05, Figure 2A) and the fungal diversity of HX (*p* < 0.05, Figure 2B) and YN (*p* < 0.05, Figure 2B) were significantly different among the compartments. However, these indices for the other three varieties of STY, SH, and HBS did not show significant difference among the compartments (Figure 2).

### 3.2. The Composition and Structure of Pummelo Mycobiota

The composition and structure of fungal communities was significantly different among the compartments (*p* < 0.001). Furthermore, in the compartments of pummelo leaf, root, and rhizosphere soil, the composition and structure of fungal communities was differently influenced by the varieties (Figure 3). Specifically, the influence of the varieties was strongest in the phyllosphere leaves (*p* = 0.003, Figure 3A), followed by the roots (*p* = 0.006, Figure 3B), but was not significant in the rhizosphere soils (Figure 3C).

The compartment effect on the composition and structure of pummelo mycobiota was consistent across all five pummelo varieties (*p* < 0.001, Figure 4A–E). Specifically, the leaf fungal communities were clearly differentiated with those of the roots and rhizosphere soils by PCoA 1, while the root and rhizosphere soil fungal communities could be differentiated by PCoA 2.

### 3.3. Variety Effect on Fungal Taxa

The pummelo leaf endophytic mycobiota were mainly dominated by the classes (Figure 5A) of Dothideomycetes (relative abundance of 53.9%), Sordariomycetes (13.8), Eurotiomycetes (6), and by the genera (Figure 5B) of *Zasmidium* (31), *Cladosporium* (7.5), and *Fusarium* (3.2). The pummelo root endophytic mycobiota were mainly dominated by the classes (Figure 5C) of Sordariomycetes (23.3), Eurotiomycetes (19.9), Dothideomycetes (15.9), and Agaricomycetes (12.7) and by the genera (Figure 5D) of *Fusarium* (8.5), *Cladosporium* (5.1), and *Mortierella* (4.9). The pummelo rhizosphere soil mycobiota were mainly dominated by the classes (Figure 5E) of Sordariomycetes (21.3), Eurotiomycetes (19.3), and Dothideomycetes (16.1) and by the genera (Figure 5F) of *Cladosporium* (7.4), *Mortierella* (5.6), *Aspergillus* (5.1), and *Fusarium* (4.5).

In the roots, the relative abundance of the top taxa did not vary significantly (Figure 5C,D). In the rhizosphere soils, the relative abundance of Eurotiomycetes ranged from 13.2 in YN to 26.8 in HBS (*p* < 0.05, Figure 5E), and the genus *Penicillium* ranged from 1.4 in SH to 7.4 in HBS (*p* < 0.01, Figure 5F). In the leaves, the relative abundance of Dothideomycetes ranged from 39.5 in SH to 69.8 in HX (*p* < 0.05, Figure 6A), and the relative abundance of Sordariomycetes ranged from 8.8 in HX to 21.7 in SH (*p* < 0.05, Figure 6A). At the genus level, the relative abundance of *Zasmidium* ranged from 20.1 in SH to 48.0 in HX (*p* = 0.06, Figure 6B), and the relative abundance of *Fusarium* ranged from 1.5 in STY to 6.9 in SH (*p* < 0.05, Figure 6B).

### 3.4. Variety Effect on Root Selection of Fungal Genera

The root selection of each fungal genus was calculated as the Log_2_ fold change [Log_2_(FC)] in the relative abundance of the genus in the roots compared to that in the rhizosphere soils. Under this criterion, the root selection pattern of the top 50 fungal genera was compared across all five pummelo varieties (Figure 7). Firstly, most fungal genera did not show a clear enrichment or depletion pattern in the pummelo roots; the selection patterns of STY, SH, and HX were more similar compared to those of HBS and YN. Secondly, the genera *Exophiala* [Log_2_(FC) > 0, *p* < 0.05 except in YN] and *Physalacria* [Log_2_(FC) > 0, not significant] were commonly enriched in the roots across the different pummelo varieties. Thirdly, the root depletion of fungal genera was dependent on the pummelo varieties; the roots of the variety YN showed strong depletion of the fungal genera *Oidiodendron* [Log_2_(FC) = −9.9, *p* < 0.01], *Arcopilus* [Log_2_(FC) = −8.9, *p* < 0.01], *Wickerhamomyces* [Log_2_(FC) = −8, *p* < 0.01], *Gliocladium* [Log_2_(FC) = −24, *p* < 0.01], *Thanatephorus* [Log_2_(FC) = −6.8, *p* < 0.01], and *Purpureocillium* [Log_2_(FC) = −25.5, *p* < 0.01].

## 4. Discussion

### 4.1. Variety Is Not a Significant Factor Affecting the Richness and Diversity of Pummelo Endophytic Mycobiota

In this study, the selected five pummelo varieties were the major cultivated varieties in the citrus production areas of Guangdong and Fujian Provinces and Guangxi Autonomous Region in China [3,30]. The varieties HX, SH, and STY are native Chinese varieties, while HBS and YN are varieties introduced from Thailand and Vietnam, respectively [1,31]. Despite their differences in origin, the observed endophytic fungal species and diversity did not vary significantly among the varieties. In a previous study, Huang et al. (2024) found that the variety was not an important factor affecting the alpha diversity of the leaf endophytic mycobiota of citrus, including mandarins, sweet oranges, pummelos, and lemons [16]. Our results are consistent with their findings: variety is a minor factor influencing the alpha diversity of pummelo microbiota, especially when compared to the factor of compartment.

When all varieties are taken together, the compartment effect was significant in influencing the endophytic mycobiota of pummelo, which aligns with the findings of studies on other plants [32,33]. However, this finding was not consistent across all five pummelo varieties. For example, the differences in endophytic fungal diversity among the leaves, roots, and rhizosphere soils were more obvious on HX and YN compared to the other varieties. These findings suggest that the compartment effect functions in a variety-dependent way; however, more experiments are needed to reveal the mechanisms of the differences among pummelo varieties.

### 4.2. Variety Effect Is Stronger on Endophytic Microbiota in the Pummelo Phyllosphere than in the Rhizosphere 

Even though the variety effect was not significant on the richness and diversity of the pummelo endophytic mycobiota, it significantly affected the composition and structure of the endophytic mycobiota of pummelo leaves and roots, but not of the rhizosphere soils. These results suggest that the pummelo varieties select their endophytic fungi with different mechanisms. For example, the plant exudates and volatile compounds are well-known for their functions in microbial recruitment and assembly [34,35]; the content and diversity of their secretion may affect the composition and structure of pummelo leaf and root endophytic mycobiota [3]. There are limited studies on the comparison of pummelo leaf and root exudates among different varieties, but Yin et al. (2023) showed that the fruit peels of different pummelo varieties were significantly different in chemical composition and functional activities [3]. The peels of STY were more abundant in volatile compounds, protein, vitamin C, and sugars and lower in organic acids than other major pummelo varieties in South China [3]. Vives-Peris et al. (2018) showed that the citrus genotypes, with larger quantities of proline and salicylates in their root exudates, positively affected the growth of rhizobacteria and further contributed to their resistance to abiotic stresses [36]. These findings suggest that the variety effect of pummelo is regulated by the different leaf and root exudates among varieties.

Furthermore, the variety effect was different among the pummelo compartments. As shown by PCoA 1 (Figure 3), 23.2% (*p* = 0.003), 10.5% (*p* = 0.006), and 10.6% (not significant) of the variation in fungal communities in the leaves, roots, and rhizosphere soils, respectively, were explained by the pummelo variety effect. Compared to the roots, the leaves are exposed to excesses of temperature, solar radiation, and moisture variations, and the leaf-secreted resources are generally more sparse and heterogeneously distributed [37,38]. Also, the microbes from the major microbial source of the leaves, airborne microbes, are more stochastic than the microbial source of the roots, namely soil microbes [39]. These factors may contribute to the greater variation in the endophytic mycobiota on pummelo leaves of different varieties.

### 4.3. The Dominant Fungi Are Differentially Distributed in Different Pummelo Varieties

The dominant fungal genera, such as *Zasmidium*, *Cladosporium*, *Fusarium*, *Aspergillus*, and *Trichoderma*, consistently occurred in the three compartments of pummelo (Figure 5B,D,F). Several species of *Zasmidium* are associated with citrus greasy spot, which is one of the most important fungal diseases infecting citrus leaves and fruits [40]. Our results suggest that the species of *Zasmidium* are important pummelo endophytes or latent pathogens not only in the leaves, but also in the roots. The species of *Cladosporium* have mainly been recorded as citrus epiphytes and saprophytes, but a recent study revealed that many of them were weak plant pathogens in the phyllosphere [41]. The species of the genera *Fusarium* and *Aspergillus* consisted of many root and stem pathogens on citrus [16]. However, the species of *Fusarium* were also found as important leaf endophytes on citrus; these species were enriched during the occurrence of citrus canker [42]. Taken together, our results suggest that the pathogen-related species are a major source of pummelo endophytes. During long-term plant–pathogen interaction, the fungal pathogens may lose their genetic virulence elements and transform to a mild endophytic lifestyle within pummelo [16]. On the contrary, the species of *Trichoderma* have been broadly recorded as biocontrol agents; their occurrence in the endophytic microbiota of pummelo may result from competitive interactions between pummelo and the microbial pathogens [43].

The variety effect on the endophytic leaf fungal communities may be associated with the different relative abundances of *Fusarium*, of Sordariomycetes, and *Zasmidium*, of Dothideomycetes, in the leaves of different pummelo varieties. Both of the fungal genera consist of abundant citrus-devastating pathogens and commensals from citrus core mycobiota [40,44], suggesting their long-term co-evolution and close relationship with plants of the genus *Citrus* [16]. The differential occurrences of *Fusarium* and *Zasmidium* in asymptomatic leaves may be associated with the co-existence and tolerance of pummelo varieties to *Fusarium* and *Zasmidium* pathogens [16]. However, the functions of most of the pummelo commensals, and also citrus commensals, are still unknown, thereby impeding our understanding of citrus–microbe interactions.

### 4.4. Compartment Effect Is Dependent on the Pummelo Variety

The compartment effect on the plant microbiome has been demonstrated with complexity on different plants [32,33]. Across our five pummelo varieties, the compartment effect was consistent in (1) shaping the composition and structure of the endophytic mycobiota in pummelo leaves, roots, and rhizosphere soils, and (2) root selection of *Exophiala* species [45]. In agriculture, several *Exophiala* species have been reported with the functions of plant growth promotion in maize [46,47] and plant pathogen suppression of Fusarium-wilt disease in strawberry [48]. The enriched *Exophiala* species in pummelo roots may function in ways similar to those in the roots of maize and strawberry, and they are worth testing in experiments.

Aside from these, the compartment effect was variety-dependent in (1) supporting the fungal richness and diversity of the endophytic mycobiota in pummelo leaves, roots, and rhizosphere soils and (2) root depletion of multiple fungal genera. The introduced varieties HBS and YN showed a distinctive fungal genera depletion pattern compared to the other three local varieties, which may be explained by their short interaction time with local soil microbes.

## 5. Conclusions

The variety and compartment effects are interactive and significant on the endophytic mycobiota of pummelos. The variety effect is significant in shaping the composition and structure of the endophytic mycobiota, the assembly of major leaf fungal taxa, and the root selection of fungal genera from the rhizosphere soils. This effect is stronger on the leaf than on the root fungal communities, and not significant on the rhizosphere soil fungal communities. The compartment effect is consistent in shaping the composition and structure of the endophytic mycobiota and root enrichment of *Exophiala* species across all varieties. However, its effect on fungal richness and diversity is dependent on the pummelo variety.

## Figures and Tables

**Figure 1 jof-12-00023-f001:**
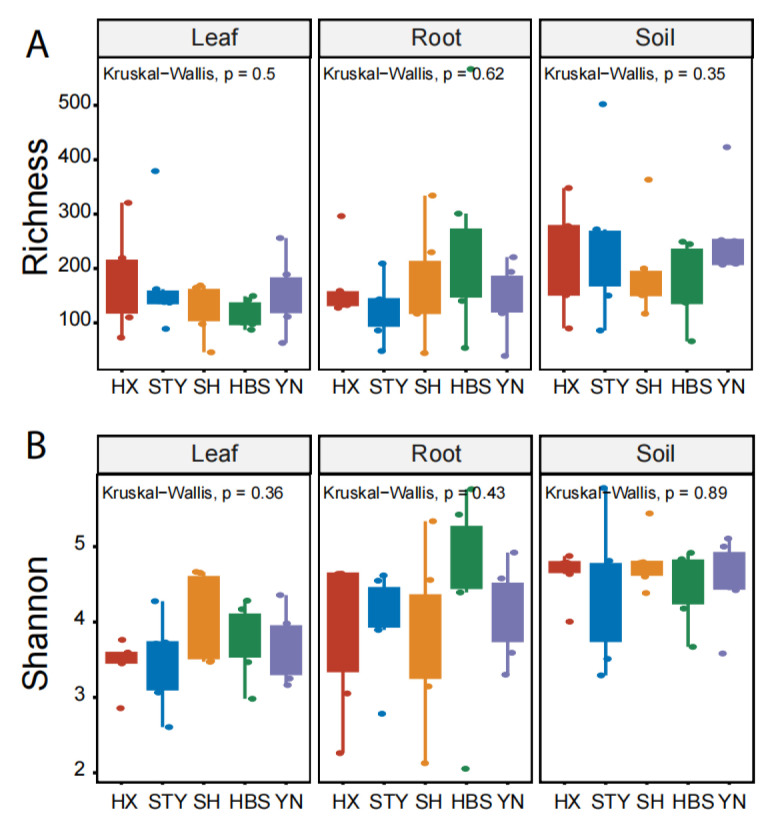
Comparison of fungal richness and Shannon diversity among pummelo varieties in the compartments of leaf, root, and rhizosphere soil. (**A**), fungal richness; (**B**), fungal Shannon diversity. Pummelo varieties: HX for Hongxinmiyou, STY for ‘Shatian’ pummelo, SH for Sanhongmiyou, HBS for ‘Siam Red Ruby’ pummelo, and YN for Vietnamese green pummelo.

**Figure 2 jof-12-00023-f002:**
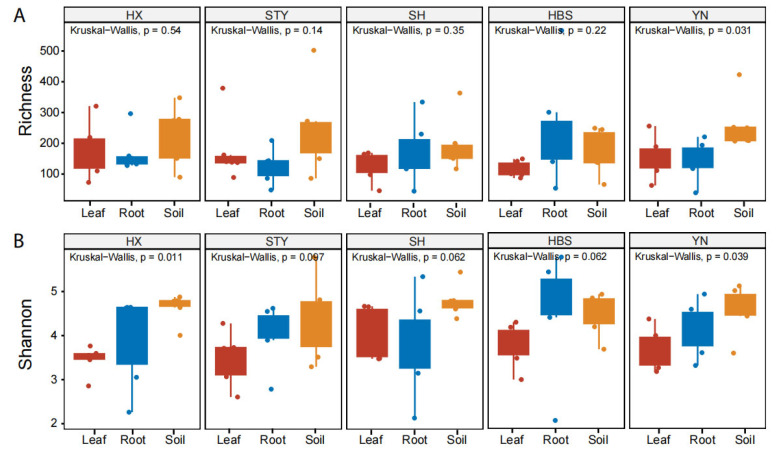
Comparison of fungal richness and Shannon diversity among pummelo tree compartments by varieties. (**A**), fungal richness; (**B**), fungal Shannon diversity.

**Figure 3 jof-12-00023-f003:**
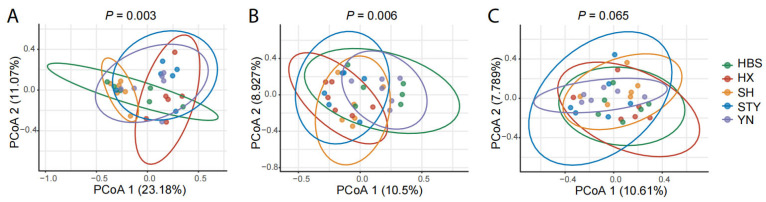
Comparison of the composition and structure of fungal communities among pummelo varieties in the compartments of leaf, root, and rhizosphere soil. (**A**), leaf fungal communities; (**B**), root fungal communities; (**C**), rhizosphere soil fungal communities.

**Figure 4 jof-12-00023-f004:**
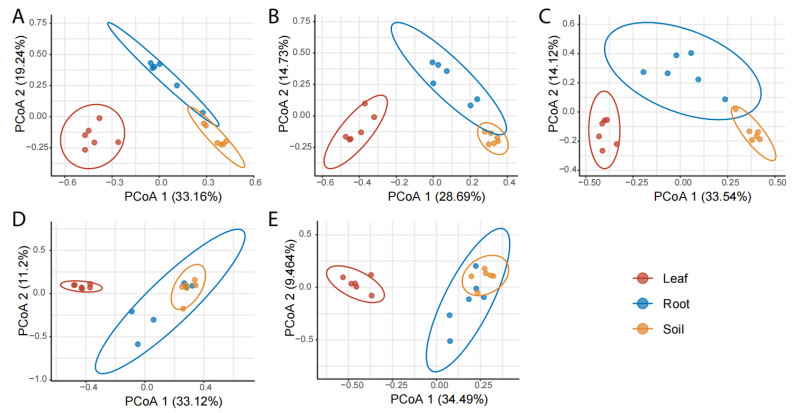
Comparison of the composition and structure of fungal communities among pummelo tree compartments by varieties. (**A**), Hongxinmiyou; (**B**), ‘Shatian’ pummelo; (**C**), Sanhongmiyou; (**D**), ‘Siam Red Ruby’ pummelo; (**E**), Vietnamese green pummelo.

**Figure 5 jof-12-00023-f005:**
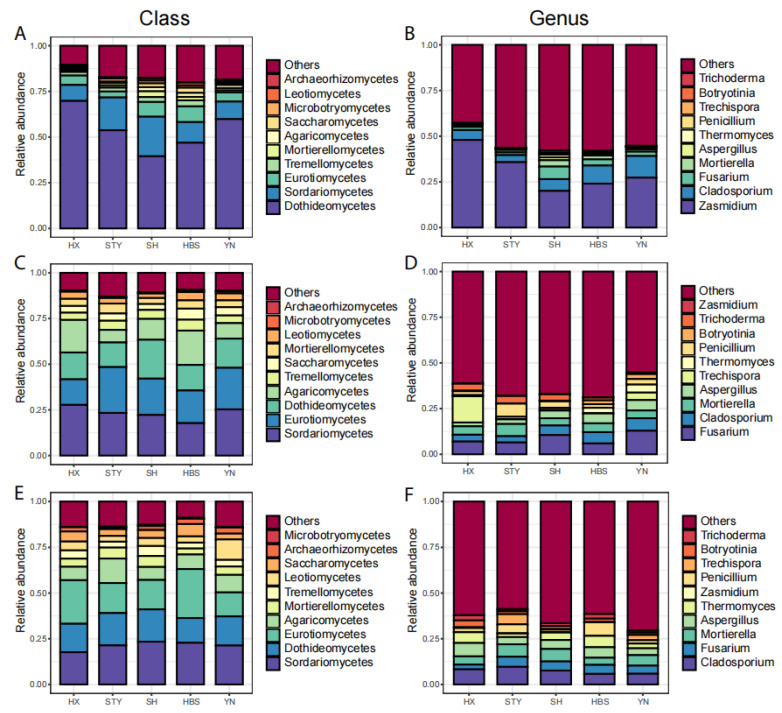
The relative abundances of top fungal classes and genera among pummelo varieties in the compartments of leaf, root, and rhizosphere soil. (**A**), leaf fungal classes; (**B**), leaf fungal genera; (**C**), root fungal classes; (**D**), root fungal genera; (**E**), rhizosphere soil fungal classes; (**F**), rhizosphere soil fungal genera.

**Figure 6 jof-12-00023-f006:**
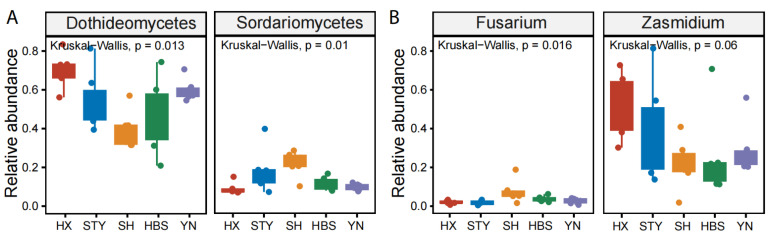
Comparison of the relative abundances of the representative fungal classes and genera among pummelo varieties in the compartment of leaf. (**A**), fungal classes Dothideomycetes and Sordariomycetes; (**B**), fungal genera *Fusarium* and *Zasmidium*.

**Figure 7 jof-12-00023-f007:**
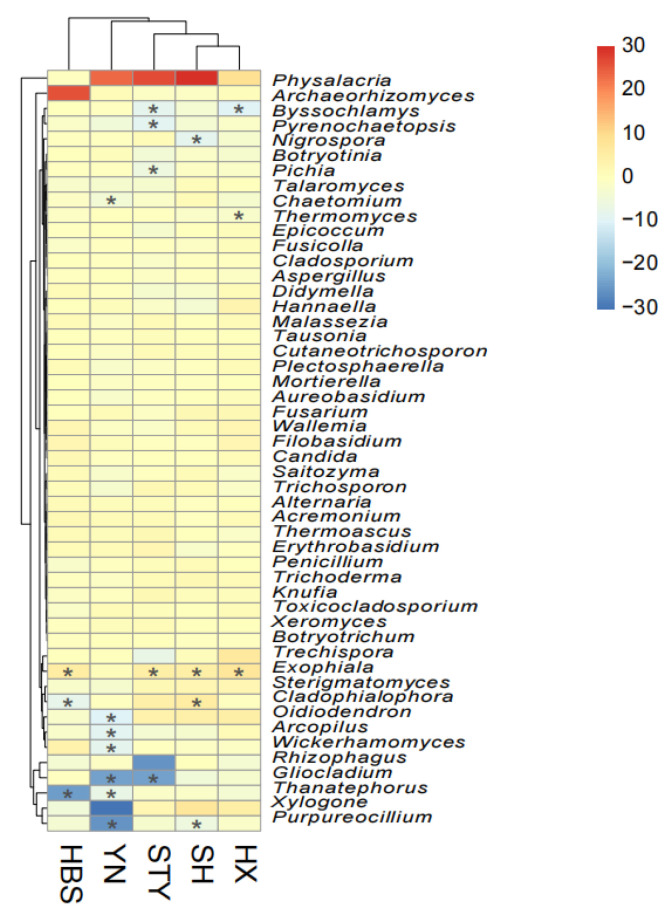
The root selection pattern of top 50 fungal genera among pummelo varieties. The grid color represents the Log_2_ fold change [Log_2_(FC)] in the relative abundance of a genus in the roots compared to that in the rhizosphere soils. “*” in the grid indicates the adjusted *p* value < 0.05.

## Data Availability

The original contributions presented in this study are included in the article. Further inquiries can be directed to the first author and corresponding author (P.W. and J.Z.). The raw sequencing data were deposited in the China National Center for Bioinformation (CNCB) with the accession number CRA035856.

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
