# Peer review of "J. Fungi2026, 12(1), 23;https://doi.org/10.3390/jof12010023"

_jof, 2025, doi:10.3390/jof12010023_

Round 1

Reviewer 1 Report

Dear authors,
Thank you for submitting this article.
This article examines the microbiome of various cultivars of the citrus pummelo plant (roots, rhizosphere, and leaves). Recent years have seen extensive research devoted to the qualitative composition and structure of the plant microbiome and its role in plant resistance to stress. Therefore, the relevance of this article is beyond doubt. The article is based on results obtained using modern molecular genetic and bioinformatics methods.
After revision, the article may be published in the journal "JoF."

Respectfully yours, reviewer.
December 8, 2025

1. The abstract needs to be revised. It is unclear, does not reflect the article's main findings, and has been poorly translated into English.
2. The article is not formatted according to the journal's guidelines. This is evident in the formatting of the references (they should be numerically listed as they are cited), the formatting of the figures should be consistent, and the formatting of the "References" section should comply with MDPI guidelines.

Author Response

This article examines the microbiome of various cultivars of the citrus pummelo plant (roots, rhizosphere, and leaves). Recent years have seen extensive research devoted to the qualitative composition and structure of the plant microbiome and its role in plant resistance to stress. Therefore, the relevance of this article is beyond doubt. The article is based on results obtained using modern molecular genetic and bioinformatics methods.
After revision, the article may be published in the journal "JoF."

Response: It is very appreciated for your time and detailed comments on our manuscript. All your comments are helpful to the improvement of the manuscript.

The abstract needs to be revised. It is unclear, does not reflect the article's main findings, and has been poorly translated into English.

Response: This part has been revised accordingly, please check it again.

The article is not formatted according to the journal's guidelines. This is evident in the formatting of the references (they should be numerically listed as they are cited), the formatting of the figures should be consistent, and the formatting of the "References" section should comply with MDPI guidelines.

Response: The Reference was edited following the MDPI guidelines.

I think the title is poorly worded and needs to be rephrased.

Response: It is changed to “The Compartment and Variety Effects Jointly Shape Pummelo Endophytic Mycobiota”. We hope you think it better this time.

In the methodological section of the paper, it would be helpful to describe the selection of pummelo varieties. While reading the article, I saw that phytopathogenic fungi were detected in the samples. Perhaps it would be worthwhile to indicate the resistance of the pummelo varieties to phytopathogenic diseases. It would be necessary to indicate whether the varieties chosen by the authors are genetically modified.

Response: Thank you for these suggestions, most of them are very interesting, and indicate there are a lot of meaningful work we should have done or we can do. Our varieties were selected because they are wildly cultivated and promoted varieties by local government. We added a sentence in the first part of M&M to describe this.

The varieties HX and STY are cultivated for a long history in China. Because they are from the same species and originated from Southeast China, they all suffer from the same plant diseases, such as citrus black spot, greasy spot, and melanose. We have not found there differences in resistance to these diseases in the field and from the literature. The varieties SH, HBS, and YN are promoted varieties, they have better taste and appearance than the other varieties. However, they have not been grown for too many years, there resistance to local pummelo diseases is not assessed systematically. We are really sorry for that we can not fulfill this suggestion in a short time.

These varieties are not genetically modified. We stressed it in the first part of M&M.

In the Discussion section, the authors write: "...differences in the diversity of endophytic fungi in leaves, roots, and rhizosphere soils were pronounced in the HX and YN varieties." However, the authors do not explain the physiological or biochemical basis for these differences. In the next paragraph of the Discussion, the authors write that the differences in the qualitative composition between the varieties may be related to plant exudates and volatile compounds, but they do not provide information on the specific substances in pummelo. It would be useful to measure and demonstrate these substances, and if this is not possible, then use literature data. I consider the explanation for the different diversity of the microbiome in the roots of plants compared to the phyllosphere insufficient. This needs to be explained in more detail. Overall, I believe the Discussion is unclear and needs more information specifically about pummelo.

Response: We really appreciate for these suggestions. They are very detailed, and offered us many information. The Discussion has been largely revised considering the suggestions and comments of both reviewers. Please check it again.

Figures must be uniform and formatted according to the journal rules.

Response: Thank you for your reminding. We adjusted the figure size accordingly. Please check them again.

Reviewer 2 Report

The study of Wu et al. is well-designed and has scientific merit. The topic is relevant for plant microbiome research, especially for crops like Pummelo. However, this manuscript has some serious shortcomings which must be improved prior to publication.

For JoF and other mdpi journals references must be numbered in order of appearance in the text and listed individually at the end of the manuscript.

Abstract

The results of mycobiota richness and diversity based on amplicon sequence variants are not summarized in the Abstract.

Key Words

I suggest excluding the word ITS from the key word list. Instead you should add “next generation sequencing”.

Introduction:

Author explained the terms phyllosphere and rhizosphere in introduction section, but the definition of “endophytic mycobiota” is missing.

Why authors use these five varieties of Pummelo (e.g., genetic distance, farmer importance…)? This should be explained when addressing the variety effect at the end of the introduction section.

Materials and methods

Data regarding the environmental factors (soil characteristics, climate condition…) and agricultural practice in orchard are missing. And since those factors are very important for microbial communities they should be included.  

Lines 115 and 116 – Write dada2 with capital letters – DADA2

Discussion

Discussion is well structured and written appropriately. But since the majority of JoF readers are mycologists, I think the ecology of fungal taxa documented in this research must be emphasized in discussion section. Authors should more discuss the most dominant genera of leaf, root, and rhizosphere mycobiota. Are some of the species belonging to these genera documented on Pummelo or on some other citruses? Are there potential pathogens? Or maybe some fungi could be beneficial for host plant.

I suggest adding few sentences regarding the ecological role of endophytes in the Discussion Section. Are they friends or foes to the host plant?

Author Response

The study of Wu et al. is well-designed and has scientific merit. The topic is relevant for plant microbiome research, especially for crops like Pummelo. However, this manuscript has some serious shortcomings which must be improved prior to publication.

Response: We really appreciate your comments and suggestions. Thank you for your time and the detailed suggestions.

For JoF and other mdpi journals references must be numbered in order of appearance in the text and listed individually at the end of the manuscript.

Response: The Reference was revised accordingly. Please check it again.

Abstract

The results of mycobiota richness and diversity based on amplicon sequence variants are not summarized in the Abstract.

Response: Done as suggested. Please check the Abstract again.

Key Words

I suggest excluding the word ITS from the key word list. Instead you should add “next generation sequencing”.

Response: Done as suggested.

Introduction:

Author explained the terms phyllosphere and rhizosphere in introduction section, but the definition of “endophytic mycobiota” is missing.

Response: We added a sentence describing “plant endophytes” in the second paragraph of Introduction, please check it again.

Why authors use these five varieties of Pummelo (e.g., genetic distance, farmer importance…)? This should be explained when addressing the variety effect at the end of the introduction section.

Response: Done as suggested. The varieties were selected based on their farming importance.

Materials and methods

Data regarding the environmental factors (soil characteristics, climate condition…) and agricultural practice in orchard are missing. And since those factors are very important for microbial communities they should be included.

Response: Done as suggested. Please check the first paragraph of this part.

Lines 115 and 116 – Write dada2 with capital letters – DADA2

Response: Done as suggested.

Discussion

Discussion is well structured and written appropriately. But since the majority of JoF readers are mycologists, I think the ecology of fungal taxa documented in this research must be emphasized in discussion section. Authors should more discuss the most dominant genera of leaf, root, and rhizosphere mycobiota. Are some of the species belonging to these genera documented on Pummelo or on some other citruses? Are there potential pathogens? Or maybe some fungi could be beneficial for host plant.

Response: We really appreciate these suggestions. We revised the Discussion according to your suggestions. Please check it again.

I suggest adding few sentences regarding the ecological role of endophytes in the Discussion Section. Are they friends or foes to the host plant?

Response: A 4.3 part was added in Discussion, please check it. Thank you again for all your suggestions, especially the suggestions on our Discussion part. These suggestions really broaden our research, the manuscript looks more abundant this time.

Round 2

Reviewer 2 Report

The Manuscript is improved now. The authors addressed very well my criticism and changed the Manuscript accordingly. To my opinion the Manuscript by Wu et al. can be now published to Journal of Fungi.

No specific comments